# The Impact of a Peer Social Support Network from the Perspective of Women with Fibromyalgia: A Qualitative Study

**DOI:** 10.3390/ijerph182312801

**Published:** 2021-12-04

**Authors:** Glòria Reig-Garcia, Cristina Bosch-Farré, Rosa Suñer-Soler, Dolors Juvinyà-Canal, Núria Pla-Vila, Rosa Noell-Boix, Esther Boix-Roqueta, Susana Mantas-Jiménez

**Affiliations:** 1Nursing Department, University of Girona, 17003 Girona, Spain; gloria.reig@udg.edu (G.R.-G.); rosa.sunyer@udg.edu (R.S.-S.); dolors.juvinya@udg.edu (D.J.-C.); suana.mantas@udg.edu (S.M.-J.); 2Health and Health Care Research Group, Department of Nursing, University of Girona, 17003 Girona, Spain; 3Institut d’Assistència Sanitària, 17244 Cassà de la Selva, Spain; nuria.pla@ias.cat (N.P.-V.); esther.boix@ias.cat (E.B.-R.); 4Nursing Department, University of Vic, 08500 Vic, Spain; rosa.noell@uvic.cat

**Keywords:** fibromyalgia, social support, long-term condition, chronic disease, social support network, qualitative study

## Abstract

Background: Fibromyalgia is a chronic and complex disease whose management by patients requires a high level of commitment. Patient empowerment therefore represents an important milestone in chronic disease treatment and control. We explored the impact of a peer social support network from the perspective of women with fibromyalgia. Methods: A generic qualitative design was proposed for the study, for which women who had been diagnosed with fibromyalgia were purposefully selected. Six semi-structured interviews were conducted, and the collected data were thematically analysed. Results: Three key themes emerged regarding the peer social support network: (1) empowerment (facilitating acceptance of the diagnosis and acting as a source of information); (2) effects on well-being and quality of life (attenuated the stigma, improved physical well-being, provided emotional support and was a socialization medium); and (3), valuable aspects (transmitted feelings of being understood and listened to and increased personal feelings of satisfaction). Conclusions: A peer social support network for women with fibromyalgia exerts positive effects on their physical, mental, and social well-being and empowers them to better manage their disease. Healthcare for women with fibromyalgia should include strategies that connect them through peer social support networks.

## 1. Introduction

Fibromyalgia (FM) is a chronic and complex disease of unknown aetiology, characterized by generalized musculoskeletal pain, whose onset and specific location is often unclear [1]. FM prevalence worldwide is 2.7%, ranging from 0.4% to 9.3% depending on the country [2]; in Spain, prevalence is 2.73% in people aged over 20 years, with a clear predominance in women (4.2% vs. 0.2% of men), and with maximum incidence among individuals aged 40–49 years (4.9%) [3].

People with FM have ongoing hypersensitivity to pain, which tends to occur mainly in muscle structures, although other symptoms and clinical manifestations affect functionality [4]. People diagnosed with FM frequently have symptoms of depression (30%), and psychological stress, anxiety, dysthymia, and simple phobia are even more frequent [5]. Emotional distress is typically accompanied by poor expectations of disease control. Other clinical manifestations of FM include altered sleep patterns, irritable bowel syndrome, fatigue, morning stiffness, memory problems, restless legs syndrome, urethritis, a subjective feeling of oedema, and cognitive impairment [6,7]. Some authors suggest that such symptoms are manifestations of central sensitization, defined as a state in which the central nervous system amplifies sensory input across organ systems, affecting perceptions of pain and producing intolerable discomfort [8,9]. The multiple symptomatology of FM is characterized by overlaps with other diagnoses, although the initial diagnosis usually focuses on inflammatory joint diseases. The complexity associated with classifying FM led the American College of Rheumatology (ACR) to establish clinical classification criteria that primarily takes into account the presence of generalized pain, the absence of any other disease explaining the pain, and the severity of symptoms [10].

Since the main problem with FM is heterogeneity, treatment needs to be multidisciplinary and personalized [7]. Pharmacological treatment combined with psychological therapy has shown clear benefits: the pharmacological treatment tackles the pain symptoms [11], whereas the psychological treatment helps manage the emotional aspects of anxiety and depression. Psychological treatments include education, cognitive-behavioural therapy, and relaxation techniques [12]. Physical exercise involving aerobic, strength, and flexibility activities has been described as an essential component of treatment, as it mitigates pain, improves mental health, and lessens the overall impact of FM on people’s lives [13]. FM-focused health programmes combining physiotherapy and cognitive-behavioural therapy have been reported to show positive results in perceptions of quality of life (QoL) for people with FM [14,15].

Management of chronic diseases like FM requires a high level of commitment from patients, so their empowerment represents an important milestone in treatment [16]. According to Nijs et al. [17], people with diffuse chronic musculoskeletal pain and poor control over FM show low tolerance levels, catastrophic thinking processes, and weak coping strategies. Several studies have shown that it easier for people to take control of chronic diseases if they are involved in groups that share knowledge and resources [18,19]. Participants in social support networks (SSNs) achieve control in less time and experience fewer complications [18,20], and this can clearly enhance QoL perceptions [18,21].

A perception of social support is the knowledge that support is available if and when needed, and is especially useful for people living with the same conditions [22]. Social support can be classified as follows: (a) information support (provision of information, guidance, advice, and recommendations); (b) tangible support (direct physical interaction); and (c), emotional support [23]. Studies have demonstrated gender-based variability in the provision and reception of social support; women tend to seek social support more than men, suggesting that women are more open and receptive to the positive benefits associated with social support [24].

SSNs are conceptualized as relationships between network participants, structure, and functionality. The relationship aspect refers to the links established between SSN participants [25]. One type of SSN is the peer support network, based on the notion that similarity between peers generates connection [26] and facilitates the sharing of information, experiences, and emotions [19]. Through SSNs, people with chronic illnesses feel encouraged to improve their coping strategies and their adherence to treatment [21,27]. An international study indicates that peer SSNs for FM facilitate social interaction and motivate patient self-care by increasing knowledge of the disease [18]. Other qualitative studies report that SSNs make valuable contributions to the battle against FM, by reconstructing identity, increasing acceptance of the disease [28], and empowering patients [29]. Furthermore, numerous physical, psychological, and social benefits were obtained from an interdisciplinary group intervention in women with FM [30]. Referring specifically to Spain, the literature on FM is scant. Existing studies only explore social support in relation to specific FM issues, such as pain [31] and sexual dysfunction [32], and no study has explored the impact of peer SSNs on FM.

We hypothesized that a peer SSN could have a beneficial effect on patients with FM. The aim of this study, therefore, was to explore the role played by peer support for people with FM, provided through an SSN created by women who had previously participated in psychoeducational groups. Exploring the impact of a peer SSN from the perspective of women with FM should enhance our understanding of the potential of SSNs so that, ultimately, we can reorient resources and adapt the healthcare offered to people with FM.

## 2. Materials and Methods

### 2.1. Design

A qualitative design based on a constructivist naturalistic approach was adopted. A qualitative methodology offers the possibility of understanding the complexity of a phenomenon from the differing points of view of informants [33]. Constructivism considers reality to be an interpretation of the world. There are as many realities as there are interpretations of a reality, and knowledge is created through common characteristics identified in all possible interpretations. Our generic qualitative design, according to Caelli et al. [34], is useful for research that—as opposed to phenomenology, fundamental theory, and ethnography—does not fall within the confines of a single established methodology [34]. Generic studies offer an opportunity for researchers to play with boundaries, use the tools provided by established methodologies, and develop research designs that fit their epistemological stance, discipline, and particular research questions [35]. The researcher using this approach can modify and adapt the study structure so that the qualitative design is tailored to the needs of the research. The focus of study is ultimately on understanding an experience or an event [33], so that we could analyse interpretations arising from the experience of participating in a peer SSN that was created from primary care-based psychoeducational groups of women with FM.

### 2.2. Study Area

The study was carried out in a Spanish primary healthcare setting that provides care to 32,600 people, namely, the Girona Health Region, where FM prevalence is 1.5%. Primary care areas, the backbone of the universal Spanish health system, are composed of one or more primary care centres that coordinate health, socio-health, and hospital services, and provide community care to people with acute and chronic health conditions. In relation to FM, in 2017, an interdisciplinary Expert FM Unit was established for the Girona Health Region to provide community and hospital care to people with FM, chronic fatigue, and chemical sensitivity. Interventions included psychoeducational groups for people with FM led by primary care nurses specializing in FM.

### 2.3. Patient Recruitment and Data Collection

The study, undertaken in February 2021, was based on semi-structured interviews with six women with confirmed FM diagnoses, who were selected using the intentional sampling [36] of patients recruited through primary care teams. The inclusion criteria was individuals aged 18 and older, who spoke Spanish or Catalan, who had been diagnosed with FM, and who were participants in a peer SSN based on previous participation in psychoeducational groups. Sociodemographic data for the patients were collected to characterize the sample.

The semi-structured, flexible, and dynamic in-depth interviews [37] aimed to capture subjective perceptions of the experience of being part of an SSN for women with FM. After a review of the literature regarding the impact of SSNs on people diagnosed with FM, a specific script was designed, and several meetings were held by the research team to develop the final set of five questions for the semi-structured guidelines (Table 1). These guidelines focused on exploring the role played by the peer SSN for women with FM, determining how the SSN affected the women’s QoL, and highlighting the most valuable social support aspects from the perspective of the participants.

The interviews were conducted with two primary care nurses based at the health centre where the interviewees received their healthcare, as it was assumed that the familiarity between nurses and interviewees would make for a more relaxed and open interview setting. According to Berger [38], familiarity may lead to the researcher having better access to and being more accepted by interviewees, whereas Hellawell [39] suggested that familiarity and empathy mixed with a sense of alienation or distance is the ideal setting for a researcher to engage with subjects.

The interviews lasted 45–60 min, according to Kvale’s [37] recommendations, took place in a comfortable, stimulus-free primary care centre room, and were audio-recorded for subsequent transcription. Interviewers used a fieldwork diary to complete information, noting and, among other issues, perceptions, sensations, and nonverbal language input.

### 2.4. Data Analysis

Two members of the research team independently analysed the transcribed interviews to establish the initially agreed categories of analysis (themes), following the six phases of thematic analysis described by Braun and Clarke [40]. The line-by-line holistic thematic analysis aimed to organize information descriptively. Open-source codes were grouped by similarity criteria, to configure the analysis categories. A subsequent phase involved theorizing about the significance of both the themes and response patterns, and interpreting how those signifiers aligned in the context of the research phenomenon to be explained [34].

The Standards for Reporting Qualitative Research (SRQR) [41] checklist was used to ensure the rigour of the research. Among other items, the researchers applied the reflexivity criterion, involving reflection on the project approach, methods, and theoretical positioning, and recording, in a fieldwork diary, personal motivations, assumptions, theoretical positions, and personal histories, leading them to pose specific research questions, as well as their analytical perspectives on exploring the data [34].

### 2.5. Ethical Considerations

The study was approved by the Ethics Committee of the Jordi Gol Primary Care Research Institute (IDIAP Jordi Gol) (reference 20/230-P), and respected the ethical principles of the most recent version of the Helsinki Declaration and good clinical practice guidelines, as established in current Spanish legislation (Law 14/2007 on Biomedical Research). All interviewees received oral and written information on the study, were guaranteed the confidentiality of their data, voluntarily agreed to participate, and provided their written consent.

## 3. Results

The six women interviewees all had a definitive FM diagnosis and were participants of a primary care-based psychoeducational group for people with FM where the peer social network was created. Mean (SD) age was 57 (15.5) years (range 42–73 years). One woman was retired, two were not working, and three were working. Five women were married or living with a partner, and one was divorced. All six had children, living with them in three cases. No major family burdens were reported, except that one woman cared for her grandchildren. Sociodemographic data on the six interviewees are summarized in Table A1 in the Appendix A.

Regarding their FM diagnosis, all six women had consulted numerous health professionals, and it had taken an average of six years to obtain a clear diagnosis. This diagnostic delay was attributed to the fact that symptoms were often blamed on psychological distress. The interviewees felt that they had not received adequate information on managing FM from healthcare professionals, and unanimously agreed that pain was a common feature of their FM.

The analysis highlighted three themes related to the role played by a peer SSN for people with FM: empowerment, effects on well-being and QoL, and valuable aspects of the SSN. Table 2 describes the topics and categories that emerged after the analysis of the interviews, whereas Table A2 in the Appendix A provides further details of the codes that emerged from the interviews.

### 3.1. Empowerment

#### 3.1.1. The SSN Facilitates Acceptance of the FM Diagnosis

Participants explained that the SSN helped them better understand their disease and that this was a key factor in accepting it.

*“The group helped me understand and accept the diagnosis.”* [P1—56 years old]

Sharing what was happening to them and what they felt meant that the women better understood the characteristics of FM, which made them feel less guilty and understand that the FM was not the fruit of their imagination or their past life.

*“What have I gained from the group? Well, understanding about this disease. What needs to be understood. That you’re not to blame.”* [P1—56 years old]

Participants placed great value on the peer SSN in helping them attribute their ailments to a specific disorder. Having a definitive diagnosis helped them to better manage their FM, and understanding its different clinical manifestations helped them better understand their disease.

*“I got a lot out of it [the group], especially the fact that I now understand that not all FMs are the same. There are people who have FM who can still work, who lead a half-normal life, there are others who do not, that the FM poses obstacles [...]. Sometimes you know, you can’t cope, you’re sinking, but then you talk to them [the group] and you know it will pass, that this is how the disease is, and that better times are coming, and that helps you.”* [P3—64 years old]

The SSN helped the women acquire a better awareness of FM. They explained that although a better awareness of the disease was very useful, they often lacked the courage to face up to it, which was why they were encouraged by the SSN.

*“I hadn’t told anyone because I thought it was something that I had to bear alone. But the group helped me understand the disease and what I had to say [to others], but as if it were a normal thing.”* [P4—73 years old].

#### 3.1.2. The SSN Is a Source of Information for Better FM Management

Regarding the sharing of doubts, experiences, information, and knowledge, the SSN was described as a valuable resource that increased the women’s knowledge of FM, resolved their doubts, helped them tackle problems, and enabled decision-making and changes in their management of FM.

*“Because they contribute things you might not have even thought about, and they make you think and see things differently”* [P6—42 years old]

The most useful information obtained from the SSN is reflected in Figure 1.

The women explained that once FM was diagnosed, they did not receive information on dealing with the disease from healthcare professionals; rather, much of this information was acquired from peers in the SSN.

*“I think the group is a positive thing, because when they [health professionals] diagnose you, they tell you what you have, but they don’t tell you what to do. They tell you to take painkillers when in pain, but they don’t tell you anything else. All this about food and relaxation has helped me a lot because the doctors don’t tell you anything.”* [P4—73 years old]

Better knowledge about FM and about lifestyles adopted by SSN peers encouraged the women to incorporate beneficial habits into their own routines and improve how they managed the disease.

*“I’ve learned the gym exercises we need to do ... I’ve also made changes in my eating habits. I’m adding things I didn’t know before.”* [P2—56 years old]

However, one perceived barrier to acquiring knowledge was people with negative perceptions participating in the SSN.

*“What happens, though, is that sometimes we meet, and someone is very negative, that makes things go wrong.”* [P1—56 years old]

### 3.2. Effects on Well-Being and QoL

#### 3.2.1. The SSN Attenuates FM Stigma

Stigma in relation to FM was identified as a crucial factor with direct negative repercussions for people with this disease.

*“For whoever has one of these diseases it’s a stigma. I don’t want it to be known. I have it, the group knows, but that’s it. I wouldn’t like it to come out.”* [P1—56 years old]

The participants explained that stigma was evident in the family and in friends, but especially in the work environment.

*“For example, I didn’t tell my friends and family until a year and a half had gone by, ... because it was a time when it was said that FM is for people who have a bit of depression that they don’t know how to handle. I didn’t mention it, not even to my children.”* [P4—73 years old]

The women’s perceptions were that they were often labelled by co-workers as people with little desire to work.

*“[They say] You’re making it ups, you’re lazy, you’re loafing, and that you don’t want to work, to do anything. And you feel very useless. And sometimes that’s what hurts you the most.”* [P2—56 years old]

FM was described as having a particular impact on strong people, who often cover up their symptoms over several years. The women considered the peer SSN to be a space where they felt liberated, as there they felt they could openly and uninhibitedly talk about the impact FM had on their lives, over and above the symptoms.

*“It’s like emptying the backpack you’re carrying.”* [P2—56 years old]

SSN participation therefore reduced both the stigma of FM and the women’s perceptions of guilt. The SSN afforded them a space where they no longer felt they had to suffer in silence but could give expression to their pain.

*“I didn’t say anything because I thought it was something I had to go through and I shouldn’t say ... I didn’t want to complain! This [group] helped me to say it, but as something normal.”* [P4—73 years old]

The women felt that the SSN enhanced their capacity to cope with their disease, to share their thoughts and feelings, and to deal with the stigma they perceived in their surroundings.

*“The group has helped me to be more honest with myself. Not to hide the disease and not to think what others might say… it has helped me to be more myself, to trust more.”* [P4—73 years old]

#### 3.2.2. The SSN Improves Physical Well-Being by Helping with Symptom Control

The women reported experiencing constant pain, which usually increased at night. Being able to share their perceptions of pain with peers in the SSN helped them control the pain; in other words, the SSN acted as a protective factor.

*“It hurts just the same, but your head is disconnected [from the pain]. And that moment of disconnection is very good for you.”* [P1—56 years old]

*“We talk a lot about the pain ... and then we send each other motivational words ... this encourages you and makes the pain less.”* [P5—49 years old]

Similarly, in relation to mood disorders, participants reported that the SSN helped them control and regulate their anxiety. Broadly speaking, the SSN was a key source of emotional support that greatly helped with mood management.

*“Uf, you know that just speaking here, I feel better!”* [P2—56 years old]

*“When someone is very down … like, today I’m feeling really depressed ...well, we encourage each other, you know.”* [P5—49 years old]

#### 3.2.3. The SSN Provides Emotional Support and Bolsters Self-Esteem

According to the participants, the peer SSN was a source especially of moral support, understood as having a purely emotional or psychological value.

*“It helps me a lot ... emotionally, to be able to get support from people you know understand you.”* [P5—49 years old]

All the women recognized the importance of this support, especially those who had less family support.

*“It’s very obvious that, for different reasons, some people do not have the support of their family and so they look for it in the group… they need this support.”* [P3—64 years old]

The women commented that the stories of their SSN peers helped them identify with others with the same disease, connected them with their own emotions, and helped them to cope better.

*“The group helped me, because of my age, I saw younger people ... I was managing well, but seeing young people with a lot of pain made me think.”* [P4—73 years old]

People with more FM-related health problems tended to have more difficulty in opening up to their SSN peers. In this case, the support was provided more individually than at the group level.

*“Some people are very down ...these we support privately.”* [P1—56 years old]

The WhatsApp group was considered to be a very powerful facilitator of exchanges of support among the SSN peers.

*“In the WhatsApp group we ask: So, how are you today? or, how are you not today?”* [P1—56 years old]

As for general emotional well-being, the participants expressed their satisfaction with the peer SSN as a resource that fostered self-esteem and positive attitudes. A more positive perspective on the disease proved to have beneficial effects on the women’s health, improved their confidence, and helped them foster healthy relationships and lifestyles.

*“There were days when I couldn’t even get dressed, it was like there was a rage inside me ... It was typical to see people exercising there [in the group] ... and with their help I was more positive, seeing that there were things that I hadn’t even thought I could do.”* [P6—42 years old]

All the participants reported that being able to help other people in the SSN, and especially those most affected by FM, had a positive impact on their own well-being, as they felt recognized, important, and necessary.

*“For example, when I explained relaxation techniques to them and saw that they wanted to know more ... I saw that what I was telling them was good for them.”* [P2—56 years old]

#### 3.2.4. The SSN Is a Socialization Medium

In relation to socializing, the women explained that the SSN increased their socialization opportunities. Participants indicated that since the SSN was created, they participated in more activities and went out more, and, as a result, they were more motivated to take care of themselves and to care for their appearance.

*“There came a time when we began to meet up even after the group ended.”* [P4—73 years old]

*“It was great sharing the experiences of the week with other people ... How did the week go? How was your week? It was that, sharing your illness with other people ... In that sense, for me it was very positive, very, very much so.”* [P3—64 years old]

The SSN made the women feel more accompanied. They explained that being part of the SSN and sharing with peers diminished their feelings of loneliness, which, in turn, had a direct and positive impact on their well-being and QoL.

*“Before finding the group…. it happens that you feel very, very alone.”* [P2—56 years old]

*“Emotionally I don’t feel so alone […] The fact of sharing and having people who understand me makes me feel less alone.”* [P5—49 years old]

Participants also reported that the peer SSN had a positive impact on their relationships with other people in their lives, including family, friends, co-workers, and other people in the community. The SSN was therefore considered to facilitate socialization, both with their SSN peers and with other people outside the SSN.

*“That helped me, to open up more… to the other people around me.”* [P4—73 years old]

### 3.3. Valuable Aspects of the SSN

#### 3.3.1. The SSN Transmits a Feeling of Being Understood

Participants reported that they felt a high degree of frustration, given that society largely does not understand what people with FM go through. They related this to feeling misunderstood and consequently not having their needs met. The SSN, they noted, helped them alleviate their feelings of frustration.

*“The group helped me accept that there were things I couldn’t do, it helped me a little, no, a lot ... because I’ve always been very active and was always doing lots of things. And feeling limited or that there were days I couldn’t even get up, that was tough … it’s like, you know, this can’t be happening to me.”* [P5—49 years old]

The SSN facilitated the women’s identification with a peer group; the group as a source of empathy meant that the women felt emotionally understood by their peers. This empathy was described as a fundamental aspect of the SSN.

*“Finding a group to share what I’m going through helped me a lot ... Before joining it, I found that many people do not understand FM [...], but here you find people that explain things and tell you things ... and it makes you feel really understood. You can share your pain or they share their pain; you understand them and they understand you [...]. You find that there are people who feel the same as you.”* [P2—56 years old]

The women explained that, in general, sharing needs, desires, perceptions, and goals with other people in the SSN increased their sense of identity with and belonging to the SSN.

*“I feel like I belong to a likeminded group of people.”* [P2—56 years old]

#### 3.3.2. The SSN Transmits a Feeling of Being Listened to

Participants reported that, in their family and social settings, they felt an absence of active listening, understood as an inability of others to know how to listen and understand. The women therefore appreciated their SSN peers for showing interest in them, making time for them, and actively listening to what they were saying.

*“We do a lot of psychology, which …. well, maybe I need it more than the others, or maybe not ... but you see people in great need, they are listened to and they feel understood.”* [P6—42 years old]

#### 3.3.3. The SSN Increases Personal Feelings of Satisfaction

An enhanced feeling of personal satisfaction was reported by the women as one of the most valued aspects of the peer SSN. They explained that, since becoming part of the SSN, they generally felt better about themselves and felt more encouraged to achieve goals in relation to their disease.

*“The talks we had (with a lot of positive emotions), the laughter … because it wasn’t just talk about FM, it was many things ... the truth is we had a great time.”* [P3—64 years old]

*“Well ... to say that it’s very positive, at least for me, who’s never been in any group ... It’s very positive .... I would 100% recommend we continue together, it’s great!”* [P6—42 years old]

## 4. Discussion

In the context of a global approach to FM, this study explored the impact of a peer SSN on women with FM who had previously participated in primary care nurse-led psychoeducational groups.

Regarding the participants, these highlighted the difficulty in obtaining a definitive diagnosis. Diagnosis is complicated by the fact that FM is often misunderstood, due to having symptoms common to certain rheumatic diseases [42]. However, obtaining a diagnosis marks a watershed in the lives of people with chronic illnesses, including FM [43], although there is a lack of evidence about how important becoming aware of FM is for patients [44].

Our findings also suggest that people with FM do not obtain sufficient information to manage their disease from health professionals; our interviewees felt that their doctors should have provided more complete information, although the information received from primary care nurses was positively appraised. Patients with FM often face a lack of understanding from doctors [45]; this is mainly because doctors typically consider the problem to be psychological [46]. We observed that most of the women started off with high expectations of the healthcare they were to receive, yet, as Chen [47] points out, people with chronic illnesses need to be encouraged to develop realistic expectations regarding healthcare.

The main themes that emerged from our analysis were empowerment, effects on well-being and QoL, and valuable aspects of the SSN. Peer SSNs make valuable contributions to empowering people with FM [29], as this empowerment increases their ability to manage their daily lives, despite the limitations imposed by their disease [48]. As for the different categories identified for the empowerment theme, diagnosis acceptance and understanding of FM heightened the women’s awareness of the disease, as corroborated by Sallinen [28], who identified peer support as an impetus to illness acceptance. Furthermore, the peer SSN was perceived to be an important source of information about FM, confirmed by García-Ríos et al. [16] and Nijs et al. [17] to be crucial to disease management. Interestingly, perceived as a barrier to the acquisition of FM knowledge through the SSN was the participation of people with negative perceptions regarding the disease. Three key aspects of the SSN were identified as especially informative. First, participants referred to relaxation techniques and tips for better sleep, corroborating the literature attesting to altered sleep patterns as a main symptom of FM [1,16]. Second, participants acknowledged the importance of physical exercise, coinciding with evidence reported by Segura-Jiménez [2] and García-Ríos [16], that physical exercise improves the pain, fatigue, and mental health problems associated with FM. Finally, diet was commented as being important, despite the lack of related evidence. People with FM often make dietary changes in an attempt to control their symptoms, with certain dietary supplements reported to improve pain symptoms [49].

As for the well-being and QoL theme, people with FM generally have a greater overall disease burden than people with other health conditions [50]. The success of a multidisciplinary nonpharmacological rehabilitation programme on QoL in patients with FM has been demonstrated by Jacobs et al. [51], whereas other authors have explored how well-being is linked to issues such as stigma and physical and mental health—categories that emerged in our study. The peer SSN in our study was perceived as a protector against stigma [52], which particularly affects the work environment [53]. As for overall health, the women in our study found that the peer SSN had positive effects on their physical, mental, and social well-being; this reflects a holistic view in line with the World Health Organization concept of health [54]. Research has confirmed the positive effects of SSNs on those three dimensions of health, e.g., Freitas and Annemas [55,56] on physical and mental health, and Prins [57] on social well-being. One important aspect of physical well-being is pain, which the women in our study confirmed as a feature of their FM, corroborating Mas et al. [4]. As for mental well-being, the women noted that the SSN fostered self-esteem and positive attitudes, both of which facilitated management of emotions and mood; this corroborated with several studies describing the same outcomes [58,59]. Mental well-being is especially important for patients with FM, given that they experience high levels of depression and anxiety [59]. Finally, participating in the SSN enhanced the women’s social well-being, through the relationships built with like-minded people; this was especially relevant, as FM often leaves people socially isolated [60], which, in turn, increases their perceptions of pain [10,57,61]. Importantly, as mentioned above, a strong thread in the analysis was that the women in our study perceived the SSN to be a protective factor for pain.

Finally, the participants in our study pointed to what they perceived as the valuable aspects of the SSN. Other qualitative studies of women with FM have reported mainly positive attributes ascribed to the support received from others with the same disease [28,62]. The women in our study reported having encountered understanding among their peers in the SSN, a quality identified as a key aspect in coping with FM [56,63]. The women attached great importance to being listened to within the SSN, and interestingly, Garcia-Campayo et al. [64] reports that being listened to by a peer group is more effective than being listened to by any other individuals. Another valued aspect of the SSN was the enhanced personal satisfaction felt by the women, derived from the feeling of being integrated in and identifying with a group of peers. Sallinen et al. [28], in their qualitative study of peer support for women with FM, have underlined the importance of feeling part of a group, and not being alone with one’s problems.

The main limitation of this study was that it explored the perceptions of a group of women with FM from the same geographical region and with a similar socioeconomic background. Furthermore, the fact that the women were recruited from psychoeducational groups may have shaped some of the findings of the study. The women who agreed to be interviewed had positive experiences with the psychoeducational programme, and the recruited women therefore had already had the opportunity to establish an SSN. The study therefore does not reflect the perspectives of women with FM who do not participate in activities or in a peer network. Other studies are therefore needed that reflect, for instance, more varied profiles. 

The results of this research highlight the key role that can be played by peer SSNs in empowering women and facilitating their management of FM. This peer support can be created through psychoeducational groups, such as nonpharmacological therapy for people with FM in our health region [65]. However, the European EULAR guidelines for nonpharmacological approaches indicate a low level of evidence for this therapy [66], whereas the NICE chronic pain guidelines make no mention of it [67]. The results of our study suggest that connecting women with FM in peer SSNs should be considered for inclusion as a care strategy, based on health professionals assessing and promoting peer SSNs in a global approach to FM, and encouraging people with FM to participate in psychoeducational groups. Further research such as RCT studies are undoubtedly needed to explore the outcomes of such strategies in the framework of health policies, and to further explore the value of peer SSNs for people living with chronic diseases.

## 5. Conclusions

A peer SSN facilitates acceptance of an FM diagnosis and is a source of information that empowers women and facilitates their management of the disease. It also exerts positive effects on their physical, mental, and social well-being. Valuable aspects of a peer SSN are that women feel understood and listened to, and feel a greater degree of personal satisfaction. Healthcare for women with FM should include strategies based on encouraging their connection through peer SSNs.

## Figures and Tables

**Figure 1 ijerph-18-12801-f001:**
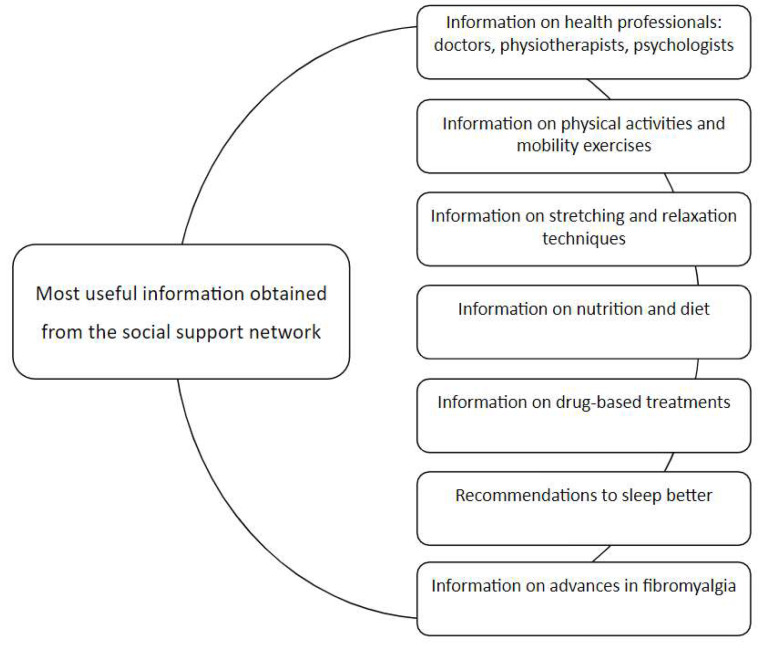
Most useful information obtained from the social support network.

**Table 1 ijerph-18-12801-t001:** Interviewing guide for people with fibromyalgia (SSN, social support network).

1. What has it meant to participate in a peer SSN?
2. What changes has participation in a peer SSN brought about?
3. What is the most valuable aspect of participating in a peer SSN?
4. How could it be explained what it represents to participate in a peer SSN?
5. Why would it be recommended to people with FM to participate in a peer SSN?

**Table 2 ijerph-18-12801-t002:** Topics and categories identified in the thematic analysis.

	TOPIC 1	TOPIC 2	TOPIC 3
	Empowerment	Effects on Well-Being and Quality of Life	Valuable Aspects of the SSN
CATEGORIES	-The SSN facilitates acceptance of the FM diagnosis	-The SSN attenuates FM stigma	-The SSN transmits a feeling of being understood
	-The SSN is a source of information for better FM management	-The SSN improves physical well-being by helping with symptom control	-The SSN transmits a feeling of being listened to
		-The SSN provides emotional support and bolsters self-esteem	-The SSN increases personal feelings of satisfaction
		-The SSN is a socialization medium	

## Data Availability

Not applicable.

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
