# Peer review of "The Impact of a Peer Social Support Network from the Perspective of Women with Fibromyalgia: A Qualitative Study"

_ijerph, 2021, doi:10.3390/ijerph182312801_

Round 1

Reviewer 1 Report

This is an valuable area of study - the impact of peer social support networks to support people with conditions that are often difficult to accept or live with. While the basis of a paper is there, as it currently stands the paper lacks a clear purpose which would greatly enhance its value. The paper needs to be framed in a broader context. For example in the Introduction it is stated this paper is different as it focuses on people in Spain in contrast to Anglo-Saxon studies. Is there an expectation and for what reasons that SSNs operate differently in Spain and how and what does this mean for the results of this study? The final paragraph of the article hints at the broader discussion in which this paper could be placed- the role of SSNs in Spanish health policies and in the Spanish health system more broadly. How does this compare to practices inn other countries or communities? 

The hypothesis of the paper is that peer SSNs could have beneficial effects on patients with FM but this taken as a given in the paper anyway - no arguments are set up to test - ie is there literature that questions the worth of the SSN and its positive impact on people's lives? The actual medical condition should just be the example.

How do the topics listed in Table 2 compare to the broader literature.

The paper includes too many quotes and care must be taken to make sure the information discussed is on topic or greater explanation is needed as to why it is included. Only use quotes that directly link to the impact of the peer SSN. For example section 3.1.1 is mostly about diagnosis which is not the responsibility of the SSN. My comment here is that this is background material that needs to be discussed when talking about the characteristics of the sample. Care also needs to taken in the distinction between the different results sections - the findings and quotes in one section could easily sit in another section. With these changes to the paper the discussion could be more of a discussion rather than a reiteration of the findings.

As a minor issue make sure that any Tables of Figures included are referred to and discussed in the text of the article.

Author Response

Dear reviewer,

We have uploaded our modified article along with the point-by-point response to your comments. Marked in green are all the changes to the article, and also the additional references.  

We are very grateful of your comments which have contributed to improve the manuscript. We now hope the article is considered suitable for publication.

Sincerely,

Gloria Reig

Reviewer 2 Report

This paper explores the impact of peer social network of fibromyalgia patients with qualitative analysis. This is an interesting study, but there are however some things that need further work.

  1. In the Introduction authors state that fibromyalgia has other manifestations such as irritable bowel syndrome or restless legs syndrome. These syndromes can co-occur with fibromyalgia, but it is still unclear whether these are separete syndromes or manifestations of same central censitization that is often stated as a central process in fibromylgia syndrome. So it is unlikely that IBD or restless legs syndrome are manifestations of fibromyalgia even though same patients often have these functional syndromes. Furhermore, I think that authors should at least mention this central sensitization hypothesis also in the Introduction section.
  2.  In The Introduction line 51 authors state that Giececke's classification is most widely used for fibromyalgia diagnosis. I don't understand this statement. At least in the litarature that I have followed (and in practice in Nordic Countries), ACR2011 or it's variations are the most videly used criteria for the diagnosis.
  3. In the Introduction section authors do not provide an overview of previous qualitative studies. There is plenty of previous evidence regarding the role of peer support.
  4. In the Materials and Methods authors refer to a generic qualitative design as a design of this study. I would like this desings of this study to be described more in detail.
  5. From Materials and Methods section it is impossible to say who were the interviewers. Did they had previous contact with the patints? What were their motives, etc. This reflection is necessary for the readers to understand what effect researchers might have had for the answers. In line 151 authors state that these issues were discussed, but I would like to see this discussion also in the manuscript.
  6. The Interview guide includes only positive or neutral staments about the participation. Was negative sides asked ar brounght up by interviews?
  7. In the Results the total number of participants does not come clear. There were six interviews, but how many participants there were in each interview?
  8. In the results section authors mention that no major family burdens were reported. Would it be unlikely that participants would like to report major family burderns in a group interview for (possibly) strangers?
  9. In Results section there is Table 2 that represents themes and authors also include quotes that illustrate differents themes. However, I would also like to see a table with different codes that emerged from the interviews and from which authors have formulated these themes. In Figure 1 there is some information from the source of information theme. That Figure is were clear and helps to understand the results.
  10. In Results section every the is associated with positive things about SSN. Did any of the participants bring out something negative? And if not why could that be?
  11. In the Discussion section some of the findins of previous qualitative studies are discussed, but I would like also to see at least a brief introduction to these themes in Introduction section.
  12. In the limitations of the study authors state that participants had similiar socioeconomic background and geoprahical area. How about time since fibromyalgia diagnosis? Age? Number of other diseases etc. It would be helpful to include a table of these characteristics of participants in the manuscript if possible. Furhermore, the lack of male participants is also a serious limitation of this study.
  13. In conclusion authors state that SSN facilitates acceptance (etc.) of FM diagnosis. However, from the study desings, we can only make assumptions of causal effects. Participants may have felt that SSN has facilitated their acceptance, but from the results I think it is only possible to make assumptions of associations. I think furher research, prefereably RCT-studies, are needed to support the statement that healtcare should include these peer SSNs. However, results of this study show that they seem to beneficial. Furthermore, there might be some bias as the role of interviwing persons is not discussed, all things that particants brounght out seem to be positive and as the authors stated, there is also at least some selection bias among participants.

Author Response

(The authors gave the same response as above.)

Round 2

Reviewer 1 Report

The changes to the paper make the research more focused now and higlight the importance of peer SSNs. It is a good little case study.

Reviewer 2 Report

Authors have made requested revisions.